# Prevalence and characteristics of childfree adults in Michigan (USA)

**Jennifer Watling Neal** **\*, Zachary P. Neal**

Psychology Department, Michigan State University, East Lansing, Michigan, United States of America

\* jneal@msu.edu

## Abstract

Childfree individuals choose not to have children, which makes them a distinctive group from parents who have had children, not-yet-parents who plan to have children, and child-less indivduals who would have liked to have children. Most research on parental status and psychosocial characteristics has not effectively distinguished childfree individuals from other non-parents or has relied on non-representative samples. In this study, we use a representative sample of 981 Michigan adults to estimate the prevalence of childfree individuals, to examine how childfree individuals differ from parents and other types of non-parents in life satisfaction, political ideology, and personality, and to examine whether childfree individuals are viewed as an outgroup. We find that over a quarter of Michigan adults identified as childfree. After controlling for demographic characteristics, we find no differences in life satisfaction and limited differences in personality traits between childfree individuals and parents, not-yet-parents, or childless individuals. However, childfree individuals were more liberal than parents, and those who have or want(ed) children felt substantially less warm toward childfree individuals than childfree individuals felt toward each other. Given the prevalence of childfree individuals, the risks of their outgroup status, and their potential role in politics as a uniquely liberal group, it is important for demographic research to distinguish the childfree from others and to better understand these individuals.

**Data Availability Statement:** Data underlying the results of our study are available from the Michigan State University Institute for Public Policy and Social Research on their website: http://ippsr.msu. edu/survey-research/state-state-survey-soss-soss-data.

## Introduction

Recent fertility rates in the United States and other Western industrialized countries are low, suggesting that many individuals are not having children [1–3]. At least some of these individuals likely identify as childfree. *Childfree* individuals voluntarily choose not to have children, and therefore potentially are quite different from individuals who also do not have children but are *not-yet-parents* or *childless*. Childfree individuals have garnered popular media attention, including discussion in an influential *Time* magazine article [4], documentaries [5], and popular press books [6, 7]. Additionally, there is growing research that aims to capture the individual characteristics of the childfree and others' attitudes toward them [8, 9].

Despite increased recognition of the childfree, research on the prevalence and psychosocial characteristics (e.g., life satisfaction, personality) of this population has been challenging for two reasons. First, much of the quantitative research on parental status has only compared

**Funding:** This study was supported by a Michigan Applied Public Policy Research (MAPPR) grant from the Institute for Public Policy and Social Research (IPPSR), and by a Departmental Collaborative Grant from Michigan State University's Psychology department.

**Competing interests:** The authors have declared that no competing interests exist.

parents to non-parents, but has not attempted to disaggregate non-parents into individuals who are childfree, not-yet-parents, and childless [9, 10]. Second, some quantitative studies that have attempted to distinguish childfree individuals from not-yet-parents or childless individuals have relied on non-representative samples [11, 12], have used definitions based on fertility rather than desire to have children [13–16], or have focused exclusively on comparisons among women [17–19].

In this study, we examine characteristics of childfree adults by asking whether individuals who identify as childfree differ from parents, not-yet-parents, and childless individuals in life satisfaction, political ideology and personality. Additionally, in view of these potential differences, we explore whether childfree individuals are viewed by others as an outgroup. The study overcomes previous challenges by employing a representative sample of Michigan residents, and by explicitly distinguishing between individuals who identify as childfree, parents, not-yet-parents, and childless. We begin by reviewing the literature on parental status and individuals' life satisfaction, political ideology, personality, and outgroup status. In the method section, we describe our data source, the *Michigan State of the State Survey*, provide our strategy for distinguishing individuals with these different parental statuses, and summarize our planned analyses. In the results section, we present descriptive information including the estimated prevalence of childfree individuals among the adult population in Michigan. We then present findings from inferential analyses examining whether the childfree differ from parents and other types of non-parents in life satisfaction, political ideology, and personality, and whether they are viewed as an outgroup. In the discussion section, we conclude by exploring the implications of our results for understanding the childfree population, identifying the limitations of our study, and suggesting future directions for research.

## Background

Childfree individuals have been recognized in the literature at least since the 1970s [20, 21] and are defined as people who do not have children and do not desire to have children in the future. Two features of this definition have made identifying and studying childfree individuals challenging. First, the *demographic* property of not having children does not distinguish childfree individuals from two other groups who also do not have children: *not-yet-parents* who plan to have children, and *childless* individuals who desired children but could not have them due to infertility or situational circumstances. Second, the *biological* property of being unable to have children (i.e. infertility), whether due to age or medical condition, is irrelevant to identification as childfree because an infertile individual who does not want children is childfree, not childless. Therefore, although demographic data typically indicate an individual's age and number of children, they provide little insight into whether an individual is childfree. This raises questions about existing estimates of the prevalence of childfree individuals in the population, and has substantially limited what is known about childfree individuals compared to individuals with other parental statuses.

### Identification and prevalence of the childfree

Most research on parental status has focused solely on distinguishing whether people have children or not. These studies are able to identify parents and non-parents, but conflate individuals who are childfree, childless, and not-yet-parents into a single category of non-parents [9, 10, 21]. Additional work has attempted to explicitly identify not-yet-parents by including questions about individuals' expectations or plans to have children in the future [22–25]. This research identifies parents, not-yet-parents, and non-parents; however, it still conflates individuals who are childfree and childless into a single category of non-parents because both of

these groups do not expect or plan to have a child, albeit for different reasons. These studies suggest that individuals who are either childfree or childless tend to have a higher education, are less likely to be married, are less likely to be religious, are less likely to have siblings, are more likely to participate in the labor force (for women) and are more likely to live in urban areas [13, 21–25].

Most of the studies that have attempted to explicitly distinguish individuals who are childfree and childless have relied on fertility data and have focused exclusively on women [13–15] or married women [16]. These studies defined women as childfree if they did not want children but were fertile or sterilized for contraceptive purposes, and defined women as childless if they had no children and were infertile for non-contraceptive reasons. Using these definitions, estimated childfree prevalence rates ranged from a low of 2.2% for married women in the 1970s [16] and to a high of 9% for women in 1995 [13], with one recent estimate placing the prevalence at 6% for women between 2006 and 2010 [15].

There are several problems with the measurement of childfree individuals in previous studies that undermine confidence in the current prevalence estimates and demographic correlates. First, much of the research continues to conflate childfree individuals with other types of non-parents, making it difficult to determine the unique prevalence and demographic correlates of this population. Second, when research has attempted to distinguish childfree individuals from other types of non-parents, it has used operational definitions that require fertility or voluntary sterilization. However, as noted earlier, fertility is irrelevant to one of the key defining features of a childfree status: a lack of desire to have children. By using a definition based on fertility, older or biologically infertile childfree women who do not desire children may have erroneously been classified as childless, thus underestimating the prevalence of childfree individuals. An operational definition based on fertility also assumes that parental status can only be achieved via biological reproduction and does not consider alternative routes to parenthood such as adoption. Third, because many of these studies focused only on women, they provide no information about the prevalence or demographic characteristics of childfree men. A more recent Pew research survey that explicitly asked women and men why they were not planning to have children estimated the prevalence of childfree individuals at a much higher rate of 23% [26].

In this paper, we address the following research question: *What percentage of the population identifies as childfree, and what are their demographic characteristics?* (**Research Question 1**). Due to problems with past measurement, we do not aim to replicate prior studies' definitions of or methods for identifying childfree individuals. Instead, to determine the contemporary prevalence rates and demographic correlates of childfree women and men in the U.S., we aim to operationalize childfree status based explicitly on individuals' *lack of desire to have children*. Because we are defining childfree status differently, we do not anticipate that the prevalence rates estimated in this study will mirror those estimated in past studies. Additionally, although we acknowledge that individuals' identified parental status can change across the life course [25], in this study we are interested in estimating prevalence rates based on individuals' current parental statuses, regardless of how they may have identified in the past or how they may identify in the future.

## Life satisfaction

The link between parental status and life satisfaction has long been of interest to researchers. On the one hand, folk and academic theories often posit that having children will increase individuals' life satisfaction and happiness [10, 27]. However, on the other hand, there is a

recognition that having children involves significant time and financial costs that might reduce life satisfaction and happiness [28].

The majority of studies that examine associations between parental status and life satisfaction have focused on cross-sectional comparisons of parents and non-parents with mixed results [10, 27, 29]. Some studies have demonstrated positive associations between parenthood and life satisfaction or happiness [30–33]. However, these findings were typically contingent on other socio-demographic characteristics. For example, some find positive associations between parenthood and life satisfaction or happiness only for men [33] or only for women [30, 31]. Additionally, although one study demonstrated positive bivariate associations between parenthood and indicators of life satisfaction or happiness [32], a re-analysis of the same data resulted in negative associations when controlling for marital status [34]. Adding to these mixed findings, additional studies demonstrated no differences in life satisfaction between parents and non-parents [35, 36], or found that parents had lower levels of happiness than non-parents [37] and that parents of school-aged children were less satisfied with their communities than others [38].

A smaller number of studies have explored differences in life satisfaction before and after the birth of a child. This design allows researchers to examine within-person changes in life satisfaction across two different parental statuses (e.g., not-yet-parent and parent). For example, research using the German Socio-Economic Panel found that life satisfaction increased during pregnancy and right after the birth of a child, but went back to original levels within two years [39]. More recently, an additional study, also using the German Socio-Economic Panel, found that having a child has led to larger within-person increases in life satisfaction among mothers since the early 2000s [40].

Although studies comparing parents to non-parents, and studies of the transition to parenthood, are common, they are unable to provide information about the life satisfaction of childfree individuals [10]. However, a handful of studies have attempted to compare different types of non-parents to parents. For example, among older Canadian adults, childless individuals were significantly less satisfied with their life than parents. However, childfree individuals were not less satisfied with their life than parents [41]. Other studies suggest that childfree individuals may experience higher levels of life satisfaction than parents and other types of non-parents [12, 17, 19, 42]. For example, some studies have shown that childfree women have expressed higher levels of global well-being than childless women [17, 42]. Furthermore, one study found that childfree women had higher levels of life satisfaction than mothers after controlling for internalized motherhood norms [19]. Finally, among both men and women, childfree individuals reported more life satisfaction than parents, although this finding was not significant when controlling for income and religion [12]. Taken together, these findings suggest a need to differentiate types of non-parents when examining associations between parental status and life satisfaction. Therefore, in this study, we ask: *Do childfree individuals differ from parents, not-yet-parents, and childless individuals in their life satisfaction?* (**Research Question 2**).

## Political ideology

Relatively little research has examined differences in political ideology by parental status. Most work has focused on broad differences between parents and non-parents, but with mixed findings. Some studies find that parents are more liberal than their non-parent counterparts. For example, focusing on a broad range of issues including education, social welfare, and defense spending, some research finds that motherhood is liberalizing [43, 44]. Adopting a narrower focus, additional work similarly finds that support for taking actions to mitigate climate

change translated into support for the center-left New Zealand Labour party, and opposition to the center-right National party, but only for parents [45]. Moreover, there is some evidence that the liberalizing effect of parenthood extends to not-yet-parents. For example, not-yet-parents held more liberal views about federal spending and women's rights than childless and childfree individuals [46].

However, other studies have reached the opposite conclusion, that parents are more conservative than their non-parent counterparts. For example, multiple studies have demonstrated a positive association between number of children and conservatism [47, 48]. This association between parenthood and conservative views has been demonstrated for specific issues as well: in the United States, parents were 15% more likely than non-parents, and very conservative individuals were 230% more likely than moderates, to believe that "vaccines should be a parent's choice" [49]. Examining differences between childfree individuals and other groups, one study found that childfree individuals were more politically liberal than parents or not-yet-parents [11]. Likewise, another study found that childfree women were more likely to view religion as unimportant, and more likely to hold egalitarian views of gender roles, than parents [13].

Much of this past work on political ideology and parental status has failed to effectively distinguish the multiple categories of non-parents (i.e. not-yet-parents, childfree, and childless). Because these groups may be ideologically heterogeneous, their conflation in past research may explain these mixed findings. Thus, in this study we ask: *Do childfree individuals differ from parents, not-yet-parents, and childless individuals in their political ideology?* (**Research Question 3**).

## Personality

Several studies, drawing on participants from many countries (i.e., U.S., U.K., Finland, Germany, and the Netherlands), have explored whether Big Five personality traits are associated with the probability of becoming a parent. Findings suggest that the probability of becoming a parent is associated with lower openness to experience [50, 51] and lower levels of neuroticism [51, 52]. Findings have been mixed for extraversion, with several studies demonstrating that becoming a parent was linked to higher extraversion [50, 51, 53] or sociability [54] and other studies demonstrating no association [55, 56]. Finally, among women only, the probability of having children has been linked to higher levels of agreeableness [50, 51, 53, 55, 57] and lower levels of conscientiousness [50, 51, 55, 57].

Fewer studies have directly examined the personality traits of childfree individuals or other types of non-parents. Among adolescents and young adults, lower levels of agreeableness and higher levels of neuroticism are linked to more ambivalence toward parenthood [58]. In addition, more ambivalence toward parenthood was linked to the postponement of having children. Additional research suggests that individuals who are explicitly childfree had lower levels of extraversion and agreeableness than parents and not-yet-parents [11]. Moreover, among childfree individuals, early articulation of a decision to be childfree was linked to higher levels of openness to experience [11]. Finally, among men, extraversion was negatively related to being childfree [59]. Extending this nascent literature, we ask: *Do childfree individuals differ from parents, not-yet-parents, or childless individuals in the Big Five personality traits?* (**Research Question 4**).

## Outgroup status

Due to the dominance of pronatal norms, the decision not to have children is often a stigmatized one, which has led childfree individuals to be viewed as an 'outgroup' toward whom

others feel less warm [8, 60, 61]. Because these norms are presumed to be stronger for women, many of these studies have focused on bias toward or stereotyping of childfree women [62–66]. For example, one study found that college students rated childfree women as less warm (*M* = 3.09, *SD* = 0.79) than mothers (*M* = 4.07, *SD* = 0.62) on a five-point scale [62]. Similarly, another study found that college students rated childfree women lower on stereotypically positive traits for women (e.g., feminine, nurturing; *M* = 5.39, *SD* = 0.77) than mothers (*M* = 5.82, *SD* = 0.67) on a seven-point scale [65]. Other studies have replicated these patterns for men and childfree couples [67–71]. For example, college students express more moral outrage toward childfree individuals (*M* = 1.37, *SD* = 0.57) than parents (*M* = 1.16, *SD* = 0.33) on a five-point scale [69], to to rate mothers (5.17) and fathers (4.62) as warmer than childfree women (4.75) and men (4.36), on a seven-point scale [67].

While most prior work has examined perceivers' differential evaluations of parent and childfree targets, one recent study used a childfree implicit association test to compare parent and childfree perceivers' evaluations of childfree individuals [72]. In a convenience sample of adults recruited via social media sites, parents had a larger IAT score (0.223) than non-parents (0.007), suggesting that parents have stronger implicit bias against the childfree than non-parents.

Although the existing literature includes robust evidence that childfree individuals are viewed as an outgroup toward which others feel less warm, these findings must be viewed in light of two caveats. First, except for one early study of randomly sampled residents from two Boston communities [70], all these studies have relied on convenience samples of undergraduate college students. Therefore, it is unknown to what extent childfree individuals are viewed as an outgroup by the general population. Second, while the observed differences in warmth between parents and childfree individuals are statistically significant, the absolute effect sizes are often modest [73]. For example, although one widely-reported study observed more moral outrage toward the childfree than parents, both groups were near the lowest reportable levels of moral outrage and differed by only 0.21 points on a 5-point moral outrage scale [69]. To better understand the impact of individuals' real or perceived differences from childfree others, we ask: *Are childfree individuals viewed as an outgroup toward whom others feel less interpersonal warmth?* (**Research Question 5**).

## Method

### Data & setting

This study uses State of the State Survey (SOSS) data collected online by YouGov, under contract with the Michigan State University Institute for Public Policy and Social Research, between May 8th and 25th, 2020. Data collection occurred during the initial wave of the COVID-19 pandemic, during a stay-at-home order in Michigan. Data collection ended on the same day that George Floyd was murdered by a police officer in Minneapolis, resulting in widespread protests against police brutality and systemic racism. Based on the survey's timing, respondents' attitudes and life satisfaction may have been affected by the COVID-19 pandemic, but were not influenced by the murder of George Floyd or subsequent protests.

### Sample

As part of the SOSS, YouGov collected data from 1,086 respondents. Using a sampling frame constructed from the 2016 American Community Survey, these respondents were matched by YouGov on gender, age, race, and education to provide a representative sample of 1,000 Michigan adults.

## Measures

**Parental status.**   We measured participants' parental status using a series of three yes/no questions:

1. Do you have, or have you ever had, any biological or adopted children?

2. Do you plan to have any biological or adopted children in the future?

3. Do you wish you had or could have biological or adopted children?

Participants who answered yes to the first question were skipped out of subsequent questions and were categorized as *parents*. Participants who answered no to the first question but yes to the second question were skipped out of the third question and were categorized as *not-yet-parents*. Participants who answered no to both the first and second question but yes to the third question were categorized as *childless*. Finally, participants who answered no to all three questions were categorized as *childfree*. This approach to identifying childfree individuals differs from prior research in two important ways. The second question allows us to distinguish individuals who expect to have children in the future (i.e. not-yet-parents) from those who do not (i.e. childless and childfree). Additionally, the third question classifies an individual as childfree or childless based solely on their lack of desire for biological or adopted children, regardless of their fertility status. In our analyses, parental status is treated as a categorical variable with childfree as the omitted or reference category.

**Life satisfaction.**   We measured life satisfaction using the five-item Satisfaction with Life Scale (SWLS) [74]. Participants rated items such as *"I am satisfied with my life"* on a 7-point scale ranging from 1 (*strongly disagree*) to 7 (*strongly agree*). We computed a scale score by averaging across the five items. This measure exhibited high internal consistency in the current sample ($\alpha = 0.91$).

**Political ideology.**   Political ideology was measured using a series of questions. First, participants were asked whether they think of themselves as a conservative, moderate, or liberal. For participants identifying as conservative or liberal, a follow-up question was used to assess whether they identified as very or somewhat conservative/liberal. For participants identifying as moderate, a follow-up question was used to assess whether they were closer to the conservative side, were in the middle, or were closer to the liberal side. The resulting responses were categorized on a 7-point scale ranging from 1 (*very conservative*) to 7 (*very liberal*).

**Personality traits.**   To measure personality traits, we used the 20-item Mini International Personality Item Pool (mini-IPIP) [75]. The mini-IPIP includes five 4-item subscales for each of the Big Five personality traits (i.e., extraversion, agreeableness, conscientiousness, neuroticism, and intellect). Extraversion is measured with items such as *"I am the life of the party"*. Agreeableness is measured with items such as *"I sympathize with others' feelings"*. Conscientiousness is measured with items such as *"I get chores done right away"*. Neuroticism is measured with items such as *"I have frequent mood swings"*. Finally, intellect is measured with items such as *"I have a vivid imagination"*. Participants rated each item on a 5-point scale ranging from 1 (*very inaccurate*) to 5 (*very accurate*). We computed subscale scores for each personality trait by averaging across the four items, reverse coding when necessary. Internal consistencies were acceptable in the current sample (extraversion $\alpha = 0.78$, agreeableness *alpha* = 0.73; conscientiousness $\alpha = 0.66$, neuroticism $\alpha = 0.72$; intellect $\alpha = 0.71$).

**Warmth toward childfree women and men.**   To evaluate whether childfree individuals are viewed by others as an outgroup, we measured participants' warmth toward childfree women and men using two separate items:

1. On a 0 to 100 scale, where 0 means very cold or unfavorable, and 100 means very warm or favorable, how warm or favorable do you feel toward women who never want to have or adopt children?

2. On a 0 to 100 scale, where 0 means very cold or unfavorable, and 100 means very warm or favorable, how warm or favorable do you feel toward men who never want to have or adopt children?

**Other covariates.**   Following the lead of other recent work on similar topics [36], we included gender, race, education, age, age$^2$, and relationship status as covariates. Gender was coded as a binary variable where 0 reflects male respondents and 1 reflects female respondents. Race was coded as a binary variable where 0 reflects respondents who identified as White alone and not Hispanic, while 1 reflects respondents who identified as a Person of Color. Education was measured as the highest degree completed by respondents and was coded as a 7-point ordinal scale: 1 (Less than High School), 2 (High School Diploma or GED), 3 (Some College) 4 (Technical College/Junior College Graduate) 5 (4 Year College Graduate), 6 (Some Graduate School) and 7 (Graduate Degree). We calculated age by subtracting respondents' answers to the question, *In what year were you born?*, from 2020 (i.e., the year the survey data were collected). Finally, relationship status was coded into three categories: *partnered* (omitted category; includes respondents identifying as 'Married or Remarried' or 'Member of an unmarried couple'), *formerly partnered* (includes respondents identifying as 'Divorced', 'Separated', or 'Widowed'), and *single* (includes respondents identifying as 'Single, never been married').

## Analytic plan

To estimate population prevalence of childfree individuals **(Research Question 1)**, we compute means and standard errors from stratified weighted survey data using the R `survey` package [76]. To explore the remaining research questions **(Research Questions 2—5)**, we estimate a series of ordinary least squares multiple regressions using unweighted survey data. We use unweighted data for these models because the sampling weights are a function of covariates already included in the models [77]. For each dependent variable, we first estimate a reduced model that includes only parental status, then estimate a full model that controls for gender, race, education, age, and relationship status. In each of these models, we use childfree as the reference group, which means that each model's estimated intercept represents the dependent variable's mean for childfree individuals, while the estimated coefficients for each of the other groups represent (and test for) the respective group's difference from the childfree.

We conducted an *a-priori* power analysis using the `pwr` package in R [78]. In our model with the smallest sample size (predicting openness, N = 876) we have 80% power at $\alpha = 0.05$ to detect an effect of $f^2 = 0.012$, which is a very small effect [79]. Because all of our models have sufficient statistical power to detect substantively meaningful effects, we interpret a lack of statistical significance in the effect of parental status as evidence of a null effect. The data and code necessary to reproduce the results reported below is available at https://osf.io/45v6b.

## Results

### Population and sample characteristics (RQ1)

Table 1 reports the population parameters estimated from stratified, weighted survey data for all Michigan adults (column 1) and for each parental status group (columns 2—5), as well as

**Table 1. Population parameters and sample descriptives (RQ1).**

| | Population parameters inferred from complete weighted data: M (SE) | | | | | Analytic Sample | |
|---|---|---|---|---|---|---|---|
| | **Population** | **Parents** | **Not-yet-parents** | **Childless** | **Childfree** | **Mean (SD)** | **N** |
| Parental Status – | | | | | | | |
| Childfree | 0.27 (0.02) | | | | X | 0.24 | 981 |
| Parent | 0.54 (0.02) | X | | | | 0.57 | 981 |
| Not-yet-parent | 0.12 (0.02) | | X | | | 0.11 | 981 |
| Childless | 0.08 (0.01) | | | X | | 0.08 | 981 |
| Relationship Status – | | | | | | | |
| Partnered | 0.51 (0.02) | 0.66 (0.03) | 0.28 (0.05) | 0.31 (0.06) | 0.35 (0.04) | 0.56 | 981 |
| Formerly Partnered | 0.17 (0.01) | 0.23 (0.02) | 0.01 (0.01) | 0.24 (0.06) | 0.11 (0.02) | 0.18 | 981 |
| Single | 0.32 (0.02) | 0.11 (0.02) | 0.71 (0.05) | 0.45 (0.07) | 0.54 (0.04) | 0.26 | 981 |
| Gender – | | | | | | | |
| Male | 0.48 (0.02) | 0.42 (0.03) | 0.59 (0.07) | 0.56 (0.07) | 0.51 (0.04) | 0.42 | 982 |
| Female | 0.52 (0.02) | 0.58 (0.03) | 0.41 (0.07) | 0.44 (0.07) | 0.49 (0.04) | 0.58 | 982 |
| Race – | | | | | | | |
| White | 0.77 (0.02) | 0.79 (0.02) | 0.7 (0.07) | 0.74 (0.06) | 0.78 (0.04) | 0.77 | 981 |
| Person of Color | 0.23 (0.02) | 0.21 (0.02) | 0.3 (0.07) | 0.26 (0.06) | 0.22 (0.04) | 0.23 | 981 |
| Continuous Variables – | | | | | | | |
| Education | 3.46 (0.08) | 3.4 (0.1) | 3.73 (0.27) | 3.9 (0.27) | 3.3 (0.16) | 3.8 (1.81) | 982 |
| Age | 49.81 (0.81) | 55.95 (0.87) | 29.06 (1.14) | 53.33 (2.62) | 45.81 (1.64) | 52.37 (17.48) | 982 |
| SWLS | 4.21 (0.07) | 4.44 (0.08) | 4.11 (0.25) | 3.66 (0.23) | 3.96 (0.13) | 4.31 (1.53) | 944 |
| Warmth toward CF Women | 66.07 (1.19) | 64.75 (1.43) | 66.21 (4.26) | 63.16 (3.4) | 69.4 (2.72) | 67.73 (29.89) | 969 |
| Warmth toward CF Men | 63.72 (1.27) | 63.39 (1.45) | 61.14 (5.58) | 58.81 (3.1) | 66.92 (2.72) | 65.36 (30.97) | 969 |
| Extraversion | 2.71 (0.04) | 2.75 (0.05) | 2.73 (0.13) | 2.62 (0.12) | 2.64 (0.07) | 2.69 (0.97) | 922 |
| Conscientiousness | 3.63 (0.03) | 3.71 (0.04) | 3.48 (0.1) | 3.43 (0.13) | 3.59 (0.07) | 3.66 (0.81) | 933 |
| Agreeableness | 3.87 (0.03) | 3.92 (0.04) | 3.82 (0.12) | 3.9 (0.1) | 3.77 (0.07) | 3.91 (0.75) | 928 |
| Openness | 3.64 (0.04) | 3.55 (0.04) | 4 (0.14) | 3.63 (0.12) | 3.67 (0.08) | 3.64 (0.82) | 876 |
| Neuroticism | 2.76 (0.03) | 2.66 (0.05) | 2.94 (0.07) | 2.82 (0.18) | 2.86 (0.06) | 2.74 (0.89) | 933 |
| Political Ideology | 4.03 (0.09) | 3.72 (0.11) | 4.58 (0.34) | 4.1 (0.24) | 4.39 (0.16) | 4.18 (2.04) | 976 |

the characteristics of our analytic sample (columns 6 & 7). We find that over a quarter of Michigan's adult population identified as childfree (0.27), which is the second most common parental status. Parents are the most common group (0.54), while smaller proportions of Michigan adults identified as not-yet-parents (0.12) or childless (0.08).

Focusing on the childfree, over half are estimated to be single (0.54), exceeding rates of singlehood among parents (0.11) and childless individuals (0.45), but far less than the rate of singlehood among not-yet-parents (0.71). About one-third of childfree individuals are partnered (0.35), matching rates of partnership among the childless (0.31) and not-yet-parents (0.28), but much lower than rates of partnership among parents (0.66). Childfree individuals tended to be younger ($\mu = 45.81$) than parents ($\mu = 55.95$) and childless individuals ($\mu = 55.95$) but older than not-yet-parents ($\mu = 29.06$). Additionally, childfree individuals were less well educated ($\mu = 3.30$) than parents ($\mu = 3.40$), not-yet-parents ($\mu = 3.73$), and childless individuals ($\mu = 3.90$). We observe no notable differences by race across any of the parental status groups.

The last two columns of Table 1 present the unweighted descriptives and listwise sample sizes for the analytic sample used in this study. The unweighted analytic sample closely mirrored Michigan's estimated population parameters on all variables, suggesting that our analytic sample is representative of the state's adult population.

## Childfree status and life satisfaction (RQ2)

Table 2 provides estimates from regression models predicting individuals' life satisfaction. At the bivariate level we find that on average childfree individuals exhibited moderate levels of life satisfaction ($a$ = 4.09, $se$ = 0.10, $p$ <.0001), and that by comparison parents exhibited statistically significantly higher life satisfaction ($b$ = 0.42, $se$ = 0.12, $p$ = 0.0004). However, after controlling for gender, education, age, and relationship status, parents' apparently higher levels life satisfaction disappeared ($b$ = 0.2, $se$ = 0.12, $p$ = 0.11). There were no significant differences in life satisfaction between childfree individuals and childless individuals or not-yet-parents in either model.

## Childfree status and political ideology (RQ3)

Table 3 provides estimates from regression models predicting individuals political ideology. At the bivariate level, we find that on average childfree individuals hold a center-left political ideology ($a$ = 4.50, $se$ = 0.13, $p$ <.0001), while by comparison parents are more conservative ($b$ = −0.58, $se$ = 0.16, $p$ = 0.0002). These findings persist, at roughly the same levels, after controlling for gender, education, age, and relationship status. There were no significant differences in political ideology between childfree individuals and childless individuals or not-yet-parents in either model.

## Childfree status and personality (RQ4)

Table 4 provides estimates from regression models predicting individuals Big Five personality traits. Across all of these models and traits, we observe relatively few differences between childfree individuals and members of other parental status groups. At the bivariate level, compared to childfree individuals, not-yet-parents appear to exhibit more openness ($b$ = 0.30, $se$ = 0.10, $p$ = 0.003), while parents appear to exhibit more agreeableness ($b$ = 0.15, $se$ = 0.06, $p$ = 0.01) and less neuroticism ($b$ = −0.18, $se$ = 0.07, $p$ = 0.01). However, all of these apparent differences disappear after controlling for gender, education, age, and relationship status. The only

**Table 2. Predictors of life satisfaction (RQ2).**

|  | Reduced Model | Full Model |
|---|---|---|
| Intercept | 4.09 (0.1)** | 4.24 (0.13)** |
| Parent | 0.42 (0.12)** | 0.2 (0.12) |
| Not Yet Parent | 0.03 (0.18) | -0.06 (0.19) |
| Childless | -0.22 (0.2) | -0.2 (0.19) |
| Female | – | 0.05 (0.1) |
| Person of color | – | -0.12 (0.11) |
| Education | – | 0.12 (0.03)** |
| Age | – | 0.01 (0)** |
| Age2 | – | 0 (0)** |
| Former | – | -0.7 (0.13)** |
| Single | – | -0.78 (0.13)** |
| R2 | 0.02 | 0.13 |
| N | 944 | 944 |

* $p < 0.05$,
** $p < 0.01$

**Table 3. Predictors of political ideology (RQ3).**

| | Reduced Model | Full Model |
|---|---|---|
| Intercept | 4.5 (0.13)** | 4.23 (0.18)** |
| Parent | -0.58 (0.16)** | -0.42 (0.16)* |
| Not Yet Parent | 0.24 (0.24) | -0.43 (0.24) |
| Childless | -0.22 (0.26) | -0.14 (0.26) |
| Female | – | 0.18 (0.13) |
| Person of color | – | 0.62 (0.15)** |
| Education | – | 0.2 (0.04)** |
| Age | – | -0.03 (0)** |
| Age2 | – | 0 (0) |
| Former | – | 0.1 (0.18) |
| Single | – | -0.21 (0.18) |
| R2 | 0.02 | 0.12 |
| N | 976 | 976 |

* $p < 0.05$,

** $p < 0.01$

statistically significant difference in the full models suggests that not-yet-parents are slightly more agreeable than childfree individuals ($b = 0.2$, $se = 0.09$, $p = 0.03$).

## Warmth toward childfree individuals (RQ5)

Table 5 provides estimates from regression models predicting individuals interpersonal warmth toward childfree women and men. Turning first to warmth toward childfree women, at the bivariate level we find that on average childfree individuals feel warm toward childfree women ($a = 73.75$, $se = 1.94$, $p < .0001$), while by comparison parents ($b = -8.16$, $se = 2.31$, $p = 0.0004$) and childless individuals ($b = -9.55$, $se = 3.90$, $p = 0.01$) felt statistically significantly

**Table 4. Predictors of personality traits (RQ4).**

| | Openness | | Conscientiousness | | Extraversion | | Agreeableness | | Neuroticism | |
|---|---|---|---|---|---|---|---|---|---|---|
| Intercept | 3.64 (0.06)** | 3.75 (0.08)** | 3.62 (0.05)** | 3.64 (0.07)** | 2.61 (0.06)** | 2.58 (0.09)** | 3.8 (0.05)** | 3.65 (0.07)** | 2.84 (0.06)** | 2.68 (0.08)** |
| Parent | -0.05 (0.07) | 0 (0.07) | 0.1 (0.06) | -0.01 (0.07) | 0.13 (0.08) | 0.11 (0.09) | 0.15 (0.06)* | 0.05 (0.06) | -0.18 (0.07)* | -0.02 (0.07) |
| Not Yet Parent | 0.3 (0.1)** | 0.21 (0.11) | -0.12 (0.1) | 0.09 (0.1) | 0.1 (0.12) | 0.12 (0.13) | 0.14 (0.09) | 0.2 (0.09)* | 0.15 (0.11) | -0.12 (0.11) |
| Childless | 0.02 (0.11) | -0.02 (0.11) | -0.05 (0.11) | -0.14 (0.11) | -0.01 (0.13) | -0.04 (0.13) | 0.1 (0.1) | 0.07 (0.1) | -0.12 (0.12) | 0.06 (0.11) |
| Female | – | -0.13 (0.06)* | – | 0.1 (0.05) | – | -0.02 (0.07) | – | 0.42 (0.05)** | – | 0.14 (0.06)* |
| Person of Color | – | -0.03 (0.07) | – | 0.07 (0.06) | – | 0.05 (0.08) | – | -0.05 (0.06) | – | -0.13 (0.07) |
| Education | – | 0.09 (0.02)** | – | 0.01 (0.01) | – | 0.02 (0.02) | – | 0.07 (0.01)** | – | -0.04 (0.02)* |
| Age | – | 0 (0)* | – | 0.01 (0)** | – | 0 (0) | – | 0.01 (0)** | – | -0.02 (0)** |
| Age2 | – | 0 (0)* | – | 0 (0) | – | 0 (0) | – | 0 (0) | – | 0 (0) |
| Former | – | 0.05 (0.08) | – | -0.04 (0.07) | – | 0.11 (0.09) | – | 0.03 (0.07) | – | 0.01 (0.08) |
| Single | – | 0.04 (0.08) | – | -0.02 (0.07) | – | 0.01 (0.09) | – | -0.17 (0.07)* | – | 0.01 (0.08) |
| R2 | 0.02 | 0.08 | 0.01 | 0.05 | 0 | 0.01 | 0.01 | 0.12 | 0.02 | 0.11 |
| N | 876 | 876 | 933 | 933 | 922 | 922 | 928 | 928 | 933 | 933 |

* $p < 0.05$,

** $p < 0.01$

**Table 5. Predictors of interpersonal warmth toward childfree individuals (RQ5).**

| | Warmth toward CF Women | | Warmth toward CF Men | |
| --- | --- | --- | --- | --- |
| | Reduced | Full | Reduced | Full |
| Intercept | 73.75 (1.94)** | 72.97 (2.73)** | 71.09 (2.01)** | 73.09 (2.84)** |
| Parent | -8.23 (2.31)** | -9.13 (2.53)** | -7.05 (2.4)** | -7.96 (2.62)** |
| Not Yet Parent | -5.55 (3.51) | -7.44 (3.77) | -7.36 (3.64)* | -10.01 (3.9)* |
| Childless | -9.55 (3.9)* | -9.32 (3.93)* | -12 (4.05)** | -11.16 (4.07)* |
| Female | – | 5.93 (1.98)** | – | 3.96 (2.05) |
| Person of color | – | 2.09 (2.33) | – | -0.14 (2.42) |
| Education | – | 1.64 (0.54)** | – | 1.64 (0.56)** |
| Age | – | -0.06 (0.07) | – | -0.14 (0.07) |
| Age2 | – | 0 (0) | – | 0 (0) |
| Former | – | -3.23 (2.7) | – | -4.1 (2.78) |
| Single | – | -3.65 (2.71) | – | -5.71 (2.8)* |
| R2 | 0.01 | 0.04 | 0.01 | 0.04 |
| N | 969 | 969 | 969 | 969 |

* $p < 0.05$,

** $p < 0.01$

cooler toward childfree women. All of these findings persist, at roughly the same levels, after controlling for gender, education, age, and relationship status.

We observe similar patterns in warmth toward childfree men: childfree individuals feel warm toward childfree men ($a$ = 71.09, $se$ = 2.01, $p$ <.0001), while by comparison all three other parental status groups felt statistically significantly cooler toward childfree men: parents ($b$ = −6.99, $se$ = 2.40, $p$ = 0.004), not-yet-parents ($b$ = −7.36, $se$ = 3.64, $p$ = 0.04), and childless individuals ($b$ = −12.00, $se$ = 4.05, $p$ = 0.003). All of these findings persist, and in the case of parents and not-yet parents are stronger, after controlling for gender, education, age, and relationship status.

## Discussion

Childfree individuals are recognized in the popular media [4, 5, 7] and academic research [8, 9] as a distinct group of non-parents who voluntarily choose not to have children. However, to date, few research studies have attempted to distinguish childfree individuals from other types of non-parents and those that do have used non-representative samples [11, 12], relied on definitions based on fertility rather than the desire to have children [13] or focused only on women [17–19]. In this study, we attempted to fill existing gaps in the literature by exploring who the childfree are in a representative sample of Michigan adults. We estimated the population prevalence and demographics of childfree individuals, then examined whether childfree individuals differ from parents, not-yet-parents, and childless individuals in terms of their life satisfaction, political ideology, and personality. We also examined whether childfree individuals are viewed as an outgroup toward whom others feel less warm.

Using a weighted representative sample of Michigan adults, we found that over a quarter (27%) of the adult population identified as childfree. Given Michigan's adult population of 7.8 million, this suggests that over 2 million Michigan adults identify as childfree and do not want children. Moreover, among the childfree, 35% are in a partnered relationship, suggesting that couples who do not want children represent an important type of family. Interestingly, the estimated population prevalence of childfree individuals in our study dramatically exceeds the

**Table 6. Estimated population prevalence of parental statuses by subgroups (Mean (SE)).**

| | Age | | Gender | | Stress about COVID-19[a] | |
|---|---|---|---|---|---|---|
| | 18-45 | 46+ | Men | Women | High | Low |
| Childfree | 0.3 (0.03) | 0.24 (0.02) | 0.29 (0.03) | 0.25 (0.02) | 0.27 (0.02) | 0.26 (0.03) |
| Parent | 0.37 (0.03) | 0.66 (0.02) | 0.48 (0.03) | 0.6 (0.02) | 0.55 (0.03) | 0.51 (0.03) |
| Not Yet Parent | 0.27 (0.03) | 0.01 (0.00) | 0.15 (0.03) | 0.1 (0.01) | 0.1 (0.02) | 0.15 (0.03) |
| Childless | 0.06 (0.02) | 0.09 (0.01) | 0.09 (0.02) | 0.06 (0.01) | 0.07 (0.01) | 0.09 (0.02) |

[a] Measured by asking "How has COVID-19 impacted how stressed or anxious you are overall," on a five-point Likert scale where "somewhat more" and "much more" were coded "high" and "about the same," "somewhat less," and "much less" were coded as "low."

estimates of 2—9% reported by earlier studies focused on women and fertility [13, 15, 16]. One possible explanation for our much higher prevalence estimate is the fact that, unlike earlier studies estimating childfree prevalence, our sample included individuals from groups who are more likely to report being childfree: individuals beyond childbearing age, men, and those who were stressed or anxious about COVID-19. To investigate this possibility, we estimated the population prevalence of parental statuses by subgroups (see Table 6). We observe very small differences in the prevalence of identification as childfree between age-, gender-, or COVID stress-based subgroups, which suggests that our higher prevalence estimate is not related to the inclusion of these groups in our sample. A second possibility is that because our measurement of parental status is not based on fertility or age, it is better able to capture previously hidden childfree individuals (e.g. infertile individuals who nonetheless identify as childfree) and thus provides a more accurate estimate of the prevalence of this identity in the population. Indeed, the prevalence rate of childfree individuals in our study is comparable to the prevalence rate in another recent survey conducted by Pew Research Center that also explicitly asked respondents whether they wanted children [26]. Although future research is needed to verify the prevalence of childfree individuals, because we find that over 1 in 4 Michigan adults identified as childfree, it is important to better understand this sizeable group of individuals.

In many ways, childfree individuals are similar to parents, not-yet-parents, and childless individuals. After controlling for demographic characteristics, we found *no differences in life satisfaction* between childfree individuals and parents, not-yet-parents, or childless individuals. This finding mirrors some past research comparing the life satisfaction of parents and childfree individuals [12, 41] and is consistent with the notion that other demographic characteristics (e.g., relationship status, education, age) are more important correlates than parental status [34]. Moreover, this finding adds to a growing body of literature that contradicts folk and academic theories that having children leads to higher levels of life satisfaction [27].

With only one exception, after controlling for demographic characteristics, we also found *no differences in personality traits* between childfree individuals and parents, not-yet-parents, or childless individuals. We do find that childfree individuals were less agreeable than not-yet-parents. However, failure to replicate other personality trait differences identified by prior research may be explained by our ability to control for a broader range of demographic characteristics than earlier studies [11].

The fact that we were unable to identify significant differences in personality traits between childfree individuals and parents suggests that the decision to be childfree is not driven by individual personality traits, but instead may be driven by other individual (e.g., political ideology) or situational (e.g., economic) factors. Indeed, we did find that childfree individuals were significantly more liberal than parents, even after controlling for demographic characteristics.

More liberal individuals may be more likely to decide to be childfree to promote or facilitate more egalitarian gender roles [13], or out of a concern for the environment [80], recognizing that choosing not to have children is the single most impactful action that an individual can take to reduce carbon emissions [81].

Although childfree individuals and couples are numerous in the population, and although they are similar in most respects to individuals with other parental statuses, our results suggest that they may still be viewed by others as an outgroup. After controlling for demographic characteristics, individuals who have or want(ed) children felt substantially less warm toward childfree individuals than childfree individuals felt toward each other. For example, childfree individuals indicated an average interpersonal warmth toward other childfree individuals of 73˚, while others felt 8-11˚ cooler toward childfree individuals. To contextualize these values and their difference, it is similar to Catholics' warm feelings toward each other (83˚) and Protestants' much cooler feelings toward Catholics (66˚) [82]. Although the difference in interpersonal warmth that we observe is modest in absolute numbers, recent reports suggest that it may have real effects, for example, in limiting childfree individuals' ability to request the same work-life balance accommodations offered to parents [83, 84].

The current study has several strengths including measurement that allowed us to distinguish childfree individuals from other types of non-parents, independently of these individuals' fertility, in a large, representative sample. However, our results should be interpreted in light of some limitations. First, although our sample was representative, it was only drawn from the state of Michigan. Notably, the state of Michigan closely resembles the overall US population in terms of race (78.2% White vs. 72%), age (median 39.8 vs. 38.4), income (median $59,584 vs. $65,712), and education (30% at least a BA vs. 33.1%). Nevertheless, future studies should examine the prevalence and characteristics of childfree individuals in a nationally representative sample. Second, our data were cross-sectional and do not allow us to look at changes in individuals' identification as childfree across the life course. Future studies should incorporate longitudinal trend and within-person panel designs to characterize trends in the prevalence of childfree individuals over time and to better understand the factors influencing the decision to be childfree. Third, although our total sample was large, small samples within parental status groups limited our statistical power to explore potential intersectionalities using parental status interactions (e.g., parental status-by-gender, parental status-by-race, or parental status-by-age interactions). Future studies with larger samples or designs that oversample individuals with less common parental statuses (i.e., childfree individuals, childless individuals and not-yet-parents) could examine these potential interactions in more depth.

Our study was conducted within the unique context of the global COVID-19 pandemic, which altered family dynamics [85] and increased unpaid care work [86]. It is possible our findings may be driven, in part, by conditions of the pandemic. However, our childfree prevalence rate mirrored pre-pandemic estimates found in a national sample using similar measurement [26] and prevalence rates did not significantly differ by reported levels of stress due to COVID-19. Furthermore, the childfree prevalence rate in our study was 24% for older respondents who, given their age, were unlikely to have changed their plans to have children due to the pandemic. Still, it would be helpful to collect post-pandemic data to examine the potential influences of COVID-19 on childfree prevalence rates as well as on interactions between parental status and the other psychosocial characteristics.

Despite these limitations, the current study adds to a growing body of literature on childfree individuals [8, 9]. Our findings indicate that the current prevalence of childfree individuals dramatically exceeds prior estimates, and that childfree individuals and couples may be more numerous in the U.S. than researchers previously thought [13, 15, 16]. Furthermore, although childfree individuals are viewed as an outgroup toward whom others feel less warm, they are

generally quite similar to parents, not-yet-parents, and childless individuals in life satisfaction and personality. Given the prevalence of this often-overlooked parental status, the risks of their outgroup status, and their potential role in politics as a uniquely liberal group, it is important for demographic research to distinguish the childfree from others and to better understand these individuals who choose not to have children.

## Author Contributions

**Conceptualization:** Jennifer Watling Neal, Zachary P. Neal.

**Data curation:** Jennifer Watling Neal, Zachary P. Neal.

**Formal analysis:** Jennifer Watling Neal, Zachary P. Neal.

**Funding acquisition:** Jennifer Watling Neal, Zachary P. Neal.

**Investigation:** Jennifer Watling Neal, Zachary P. Neal.

**Methodology:** Jennifer Watling Neal, Zachary P. Neal.

**Writing – original draft:** Jennifer Watling Neal, Zachary P. Neal.

**Writing – review & editing:** Jennifer Watling Neal, Zachary P. Neal.

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
