## [Decision Letter · Decision Letter 0]

11 May 2021

PONE-D-21-11425

Who are the Childfree?

PLOS ONE

Dear Dr. Jennifer,

Thank you for submitting your manuscript to PLOS ONE. After careful consideration, we feel that it has merit but does not fully meet PLOS ONE’s publication criteria as it currently stands. Therefore, we invite you to submit a revised version of the manuscript that addresses the points raised during the review process.

Dear Author(s),

Thank you for sending your article for consideration. It is important and interesting. There is also an opportunity to expand the research by considering others who are with a child.

Based on the advice I have received, I ask that you follow the comments and suggestions of the reviewers.

We look forward to receiving your revised version soon.

Yours sincerely,

We look forward to receiving your revised manuscript.

Kind regards,

Shah Md Atiqul Haq

Academic Editor

PLOS ONE

Journal Requirements:

3. Please modify the title to ensure that it is meeting PLOS’ guidelines (https://journals.plos.org/plosone/s/submission-guidelines#loc-title). In particular, the title should be "specific, descriptive, concise, and comprehensible to readers outside the field" and in this case it is not informative and specific about your study's scope and methodology.

Reviewers' comments:

Reviewer's Responses to Questions

**Comments to the Author**

1. Is the manuscript technically sound, and do the data support the conclusions?

Reviewer #1: Partly

Reviewer #2: Yes

Reviewer #3: Yes

2. Has the statistical analysis been performed appropriately and rigorously? 

Reviewer #1: Yes

Reviewer #2: Yes

Reviewer #3: Yes

3. Have the authors made all data underlying the findings in their manuscript fully available?

Reviewer #1: Yes

Reviewer #2: Yes

Reviewer #3: Yes

4. Is the manuscript presented in an intelligible fashion and written in standard English?

Reviewer #1: Yes

Reviewer #2: Yes

Reviewer #3: Yes

5. Review Comments to the Author

Reviewer #1: This article investigates the correlates of attitudes about being childfree (life satisfaction, political ideology, personality, and warmth toward childfree individuals), among which the authors find that there are virtually no differences between childfree individuals and those with children, except for political ideology and warmth toward childfree individuals. However, a caveat that merits attention is the very low R2 in all of the models, ranging from 0.04 to 0.13 in the full models (and even lower in their reduced models). This means that, despite their statistical significance, an enormous amount of variation in life satisfaction, political ideology, and warmth toward childfree individuals are going unexplained by the childfree status. This might suggest that being childfree by itself is not as important as the study believes it to be, that the models may require additional variables, and/or merits some commentary from the authors.

Another caveat that merits some attention is the fact that the one survey the authors based their study on was conducted during COVID-19. There thus appears to be some disconnection between the set-up (the literature review, whose research was done in pre-COVID-19 times) and the study (strictly during COVID-19). The authors could amend this connection by better framing the article with respect to COVID-19 and by citing more of the emerging literature on family dynamics amid COVID-19, or by analyzing longitudinal data from before the COVID-19 pandemic (to assess what things were like in the era that the article's cited research are based).

Please be mindful of spelling and grammatical errors that surface in the abstract and text.

The authors might consider placing their article in discussion with the following articles, given their research on family dynamics in COVID-19:

Lebow, J. L. (2020). Family in the age of COVID‐19. Family process.

Power, K. (2020). The COVID-19 pandemic has increased the care burden of women and families. Sustainability: Science, Practice and Policy, 16(1), 67-73.

Thank you for the opportunity to review this manuscript. I hope these comments are helpful. Good luck to the authors.

Reviewer #2: The study presents a granular overview of the childfree demographic by disaggregating non-parents into individuals who are childless, not-yet-parents, and finally those who have chosen to be childfree. This was achieved with a simple but elegant set of three questions baked into the survey. In my view, this representative survey, albeit restricted to a single region in the US, is the key contribution of the study. It shows that this specific cohort seems to account to a quarter of the US population.

The study is methodologically sound, but the authors could have further advanced the contributions of their study with more sophisticated modeling. The actual analyses performed in the paper are rather exploratory and restricted to estimating a series of ordinary least squares multiple regression.

But that doesn't detract from the paper and the presentation of what is seemingly a representative survey of childfree individuals and their counterparts. My recommendation for future research is to go back to the analysis and further explore the data. I can think of a number of questions that could be probed by deploying further survey waves to collect longitudinal data, and I'm sure the author(s) are similarly making inroads in this direction.

The paper is immaculately written despite the occasional typo (e.g., "but must lower than").

Reviewer #3: This is a straightforward article which provides a thorough analysis of 'childfree' individuals in one US state. Therefore I have only minor criticisms.

p. 4: explicate that the National party in New Zealand is the conservative party.

pp. 4-5: The claim that suspicion of vaccination is "conservative" seems to be overgeneralising from an idiosyncratic feature of the current American political scene. Is it true, for example, that in France, those who are skeptical of vaccines are more right-wing?

There is a typographical error in line 362.

Reference 4 omits necessary bibliographic information and the reference to the "Times" in the text is ambiguous—specify NYT, Times of London, or whatever.

6. PLOS authors have the option to publish the peer review history of their article (what does this mean?). If published, this will include your full peer review and any attached files.

Reviewer #1: No

Reviewer #2: No

Reviewer #3: No

---

## [Editor Report · Decision Letter 1]

18 May 2021

Prevalence and characteristics of childfree adults in Michigan (USA)

PONE-D-21-11425R1

Dear Dr. Jennifer,

We’re pleased to inform you that your manuscript has been judged scientifically suitable for publication and will be formally accepted for publication once it meets all outstanding technical requirements.

Kind regards,

Shah Md Atiqul Haq

Academic Editor

PLOS ONE

Additional Editor Comments (optional):

Dear Author.

First of all, I would like to thank you for revising the paper in response to the reviewers' comments and suggestions.

I thank you for responding to my comments and those of the reviewers.

I recommend that the paper be accepted for publication.

Best regards.
---

## [Editor Report · Acceptance letter]

27 May 2021

PONE-D-21-11425R1 

Prevalence and characteristics of childfree adults in Michigan (USA)  

Dear Dr. Watling Neal:

I'm pleased to inform you that your manuscript has been deemed suitable for publication in PLOS ONE. Congratulations! Your manuscript is now with our production department. 

Kind regards, 

on behalf of

Dr. Shah Md Atiqul Haq 

Academic Editor

PLOS ONE